# Influence of Artificial Intelligence in Education on Adolescents’ Social Adaptability: A Machine Learning Study

**DOI:** 10.3390/ijerph19137890

**Published:** 2022-06-27

**Authors:** Chuyin Xie, Minhua Ruan, Ping Lin, Zheng Wang, Tinghong Lai, Ying Xie, Shimin Fu, Hong Lu

**Affiliations:** 1School of Education, Guangzhou University, Guangzhou 510006, China; crazy.xiaoxun@163.com (C.X.); r13416170138@126.com (M.R.); lin_ping678@163.com (P.L.); xttya@126.com (T.L.); fusm@gzhu.edu.cn (S.F.); 2Management Center for Quality Education of Baiyun District, Guangzhou 510080, China; wangzheng65@126.com; 3School of Nursing, Gannan Medical University, Ganzhou 341004, China; 4School of Public Administration, Guangzhou University, Guangzhou 510006, China

**Keywords:** artificial intelligence in education, adolescent, social adaptability, machine learning

## Abstract

This study aimed to investigate the influence of artificial intelligence in education (AIEd) on adolescents’ social adaptability, as well as to identify the relevant psychosocial factors that can predict adolescents’ social adaptability. A total of 1328 participants (mean_age_ = *13.89*, SD = *2.22*) completed the survey. A machine-learning algorithm was used to find out whether AIEd may influence adolescents’ social adaptability as well as the relevant psychosocial variables, such as teacher–student relations, peer relations, interparental relations, and loneliness that may be significantly related to social adaptability. Results showed that it has a positive influence of AIEd on adolescents’ social adaptability. In addition, the four most important factors in the prediction of social adaptability among *AI* group students are interpersonal relationships, peer relations, academic emotion, and loneliness. A high level of interpersonal relationships and peer relations can predict a high level of social adaptability among the AI group students, while a high level of academic emotion and loneliness can predict a low level of social adaptability. Overall, the findings highlight the need to focus interventions according to the relation between these psychosocial factors and social adaptability in order to increase the positive influence of AIEd and promote the development of social adaptability.

## 1. Introduction

With the rapid development of information technology, the use of electronic devices in education becomes prevalent. For example, it is easy to find the shift from paper testing to computer-based or online testing in recent decades. In recent years, artificial intelligence, a new technology developed on the basis of information technology, has also been applied in education. However, artificial intelligence in education (AIEd) has its advantages and disadvantages. Some studies consider that AIEd brings more opportunities than threats [1,2]. Ma and her colleague [3] found that compared with traditional learning tools, intelligent tutoring systems (ITS) are more effective tools for learning by analyzing 107 studies. Erdemir and İngeç [4] also found that intelligent learning environments created through web-based tutoring systems have a positive influence on academic achievement and permanence in education. What is more, one study shows that AI can help to detect students’ emotions during class, and teachers may adjust their teaching accordingly [5].

However, the prevalent utilization of artificial intelligence in the teaching–learning process may also bring a series of disadvantages. The ethical problems of using AI to collect educational data and conduct relevant analytics have also been the focus of many studies [6,7,8,9]. In addition, Holstein [10] pointed out that the need of teachers and students toward AIEd remains unclear [10,11]. Zanetti, Iseppi, and Cassese [12] also raised their apprehension that AIEd may be “deviate” and become potentially malicious, due to programmers’ biases or other purposeful actions. Therefore, it is worth discussing the potential influence of AIEd.

### 1.1. Artificial Intelligence and Social Adaptation

AIEd may change the way of communication between teachers and students. Traditionally, teachers and students used to have face-to-face communication in a specific space and time. The emergence of AIEd may reduce the opportunities for face-to-face communication. In addition, AIEd cannot be conducted without the Internet and electronic devices. A previous study demonstrates that frequent usage of electronic devices has a negative influence on adolescents’ interpersonal relationships [13] as well as social adaptability [14]. Xie [15] also found a causal relationship between network usage and adolescents’ social adaptability. However, the issue that whether AIEd may also have a negative influence on adolescents’ social adaptability remains unclear.

On the contrary, some studies even found that AI can promote the development of adolescents’ abilities [16]. For example, Ali, Park, and Breazeal’s study [16] demonstrated that children who interact with social robots express a high level of creativity. In addition, empirical research also demonstrated that a wearable robot may enhance the expression ability of teenagers with autism spectrum disorder [17].

Adolescents are the principal recipients of AIEd and they are in a state that is very easily affected by the external environment [18,19,20]. However, previous studies only discussed the influence of AIEd on adolescents at the theoretical level, which may lead to some limitations. Considering the bidirectional influence of AIEd, we put forward:

**Hypothesis** **1.**
*AIEd has an impact on adolescents’ social adaptability.*


### 1.2. Artificial Intelligence in Education and Social Factors That Related to Social Adaptation

Ecological theory, as proposed by Brofenbrenner [21], demonstrates that both contextual and individual factors have a significant influence on adolescent development. AIEd can be regarded as contextual factors that may influence adolescents’ social adaptability while some psychosocial factors may also play crucial roles in shaping adolescents’ social adaptability. Therefore, the relevant psychosocial factors should also be taken into consideration.

AIEd may not only change the way of communication but also the relationship between teachers and students as well as peer relationships. Previous studies suggested that a good teacher–student relationship can promote the school adaptability of adolescents [22]. In addition, peer relationships also play an important role in predicting adolescents’ social adaptability. A good peer relationship can predict excellent emotional expression ability and social adaptability [23] while poor peer relationships may have a negative impact on adolescents’ social adaptability [24]. Thus, in the present study, we put forward the following hypotheses:

**Hypothesis** **2.**
*Interpersonal relations, including teacher–student relations and peer relationships are potential predictors of the status of adolescents’ social adaptability.*


**Hypothesis** **3.**
*There are significant differences in teacher–student relations and peer relationships between the AI group and the non-AI group.*


### 1.3. Personality Factors and Learner-Related Factors That Related to Social Adaptation

According to ecological theory, individual factors also play important role in adolescent development [21]. For students, individual factors can be divided into personality factors and learner-related factors.

Wohn and LaRose’s [25] study shows that loneliness, a personality factor, is an effective predictor of social adaptability. Tao and his colleagues [26] also found that social adaptability can be improved by reducing loneliness. In other words, loneliness may be closely related to the status of social adaptability among adolescents. Moreover, previous studies also found that AI agents can help patients reduce loneliness by providing social support [27]. Impulsivity, an important personality trait, is also an effective predictor of social adaptability (Moschetta, Valente, and K., 2013). The result of Valente’s research demonstrates that a high level of impulsivity may predict poor social adaptability among adolescents [28]. In addition, through analyzing 40 studies, Li and his colleagues [29] found that college students with mobile phone addiction were more likely to develop a high level of impulsivity. *Therefore, the current study* put forward the following hypotheses:

**Hypothesis** **4.**
*Personality traits, including loneliness and impulsivity, are potential predictors of the status of adolescents’ social adaptability.*


**Hypothesis** **5.**
*There are significant differences in loneliness and impulsivity between the AI group and the non-AI group.*


What is more, AIEd changes the way of teaching and learning, which may also influence students’ academic emotions. Academic emotion, an important learner-related factor that refers to various emotional experiences in connection with students’ academic activities in the learning process, was founded that plays a significant role in students’ development and is closely related to adolescents’ adaptability and problematic behavior [30]. In addition, Wang and her colleague [31] found that adolescents’ emotion regulation strategies can effectively predict their social adaptability. Thus, we put forward the following hypotheses:

**Hypothesis** **6.**
*Learner-related factors, including academic emotion and emotion regulation strategies, are potential predictors of the status of adolescents’ social adaptability.*


**Hypothesis** **7.**
*There are significant differences in academic emotion and emotion regulation strategies between the AI group and the non-AI group.*


## 2. Materials and Methods

### 2.1. Participants

The participants in this study were recruited from *13* AI demonstration schools in Guangdong province, southern China, using random sampling. These 13 schools have AI classes with AI teaching experience and non-AI classes without AI teaching experience, and the AI classes of these 13 schools have AI teaching experience for one year or more. A total of 1338 participants participated in the study. People who meet the following criteria are not eligible for the study: unable to understand the terms in the questionnaire and leaving more than 30% of items uncompleted. In addition, the missing values of the included data will be replaced with averages. Therefore, sample comprised a total of 1328 adolescents (53.01% male), ranging in age from 8 to 19 years (mean_age_ = 13.89, SD = 2.22), giving an effective return rate of 99.25%. Before completing the survey, all the participants gave written informed consent. Then, both AI group and non-AI group are required to complete the questionnaires at one time within 40 min.

### 2.2. Data Collection Instruments

*Artificial Intelligence Usage Questionnaire*: The AI questionnaire was used to investigate the usage of AI among adolescents, referring to the Mobile Phone Use Questionnaire written by Wang and his teammate [31]. The AI questionnaire consists of three dimensions, including the frequency of usage, the intention and attitude toward using AI, and the feelings after using AI. The items of the frequency of usage include “*Whether AI is used for learning*”, “*How many days do you spend on learning with AI per week*”, and “*How many hours do you spend on learning with AI every day*”. The items of the intention and attitude toward using AI include “*Why do you use AI for learning*”; “*Attitude of your family towards you learning with AI*”. The items of the feeling after using include “*You think it is helpful for you to use AI for learning*”, “*You find yourself spending more and more time on AIEd*”, “*You will feel insecure and anxious if you study without AI*”, and “*How interesting do you think it is to learn with AI*”. There are 9 items in total.

*Social Adaptability*: This scale, which was adopted from Zheng [32] includes 20 items that measure five dimensions of social adaptation, including peer relationships, self-management, academic performance, obedience, and willingness to express. Then, participants should respond “*agree*” by choosing “1” or “*uncertain*” by choosing “2” or “*disagree*” by choosing “3”. The questionnaire had high validity and reliability in this study (Cronbach’s alpha = 0.80).

*Interpersonal Relationship*: We adopted the 28 items from Zheng [32] to measure interpersonal relationships. One sample was that “*I can’t concentrate on listening to others*”. Then, participants should respond “*yes*” or “*no*” to each item. The scale had high validity and reliability in this study (Cronbach’s α = 0.88).

*Interparental Relation*: A 9-item scale was adopted to measure adolescents’ perception of interparental relations [33]. To complete this subscale, the participants were asked to answer 9 items that concerned their perceptions of interparental relations in their families. Students should respond to each item from 1 (strongly disagree) to 5 (strongly agree). A total score ranges from 9 to 45 with higher scores indicating closer relation. The scale had high validity and reliability in this study (Cronbach’s alpha = 0.92).

*Teacher–student Relation*: We adopted the 7-item scale to measure teacher–student relation [34]. One sample is “*Teacher always plays favorites*”. Students were required to respond from 1 (strongly disagree) to 5 (strongly agree). A total score ranges from 7 to 35 with higher scores indicating closer relation. The reliability and validity of the scale have been well documented (Cronbach’s alpha = 0.91).

*Peer Relation*: Adolescents’ peer relation was measured using the Chinese version of the Peer Relation Questionnaire [35](Chen and Zhu, 1997). All items were assessed using a six-point scale (1 = “strongly disagree,” 6 = “strongly agree”). For each participant, his score for all 18 items was determined, with higher scores showing higher levels of peer relationships. For the current study, the measure demonstrated excellent reliability (Cronbach’s alpha = 0.83).

*Loneliness*: We used the UCLA Loneliness Scale [36] to assess participants’ loneliness. Participants were required to rate on a 4-point Likert scale. The 4-point scale was ranging from 1 (strongly disagree) to 4 (strongly agree). Internal consistency in the current sample was adequate (α = 0.91).

*Impulsivity*: S-UPPS-P scale [37] is a 20-item scale that measures five dimensions of impulsivity, including negative urgency, positive urgency, programmatic, perseverance and sensation seeking. Each item was rated on a scale from 1 (strongly disagree) to 4 (strongly agree). The scale had suitable validity and reliability in this study (Cronbach’s alpha = 0.65).

*Academic Emotion*: A 18-item Academic Emotions Questionnaire [38] was adopted to measure two dimensions of social adaptation. All 18 items were rated using a five-point scale (1 = “strongly disagree,” 5 = “strongly agree”). For each participant, his total score for all 18 items was determined, with higher scores indicating higher academic anxiety and academic boredom. For the current study, the measure demonstrated good reliability (α = 0.92).

*Emotion Regulation Strategies*: Adolescents’ emotional regulation strategy was measured using the Chinese version of the Emotion Regulation Strategies Questionnaire [39]. A total of 10 items were assessed using a seven-point scale (1 = “strongly disagree,” 7 = “strongly agree”). For the current study, the measure demonstrated high reliability (Cronbach’s alpha = 0.86 for cognitive reappraisal sub-scale; Cronbach’s alpha = 0.64 for expression inhibition sub-scale).

*Empathy*: Basic Empathy Scale [40] was used to measure the variable of empathy in this study. There are 20 items on the scale, including two dimensions: cognitive empathy and emotional empathy. All items were rated using a five-point scale (1 = “strongly disagree,” 5 = “strongly agree”). For each participant, their total score for all 20 items was determined. The higher the score, the stronger the empathy. In this study, the measure demonstrated good reliability (α = 0.83).

This study is exploratory, and machine learning models are more suitable for exploratory research than traditional regression models. There are three factors contributing to our decision that utilize machine learning to give a deep analysis. First of all, utilizing random forest (RF) to conduct a regression model can maintain the accuracy of the study for RF is insensible to missing data. In addition, RF can also demonstrate the importance of different potential predictors. What is more, RF can also better handle the analysis of multiple variables. Therefore, to assess the difference in social adaptability between AI group and non-AI group, the partial dependence plot (PDP), a machine learning method, is employed to make a preliminary analysis. Partial dependence is a library for visualizing input–output relationships of machine learning models, which can measure the prediction change when changing one or more input features. However, PDP is plotted out based on machine learning model. In this study, RF method was also adopted. The RF method provides an ensemble learning method for classification, operates by constructing numerous decision trees, and produces the best result of classification based on the combination of individual trees. Random decision forests are able to correct the habit of decision trees overfitting to their training set.

The following is a description of the RF method executed with the Python Sklearn classification method.

Given a training set X ={x1,…, xn} with responses Y= {y1,…, yn}, random samples are selected (B times), with their replacements from the training set, and are used to train the decision trees:

For b = 1,…, B:

Sample, with replacement, B training examples from {X, Y}; call these {Xb, Yb}.

Train regression tree fb on {Xb, Yb}.

After training, a prediction for unseen sample x′ can be made by averaging the predictions from all the trained individual regression trees on x:$$\hat{f}=\frac{1}{B}\sum_{b=1}^{B}f_{b}{(x}^{\prime })$$

Based on RF model, we can conduct PDP by:$${\hat{f}}_{s}(x_{s})=E_{x_{c}}[\hat{f}(x_{s},X_{c})]=ʃ\hat{f}\left((x_{s},X_{c})\mathrm{dP}(X_{c}\right)$$

## 3. Result and Discussion

### 3.1. Artificial Intelligence Usage and Social Adaptability

The subjects are divided into two groups: the *AI* group and the *non-AI* group.

The *AI* group refers to applying AI to teaching, including learning with intelligent devices (tablet teaching, 3D printing, and the use of intelligent equipment for learning (flat teaching, 3D printing)), learning relevant courses, and carrying out relevant interest classes (programming UAV course and assembling robot), intelligent classroom/smart campus construction, etc.

The *non-AI* group refers to the group of students who adopting traditional teaching method.

As is shown in Table 1, a total of 1328 effective subjects were collected in this study, including 1016 subjects in the *AI* group and 312 subjects in the *non-AI* group.

As shown in Figure 1, we can easily find that grade, family income, age, and AIEd both have an influence on adolescents’ social adaptability. Therefore, we control these variables in this study. Then, we further analyzed the influence of AIEd on adolescents’ social adaptability. According to the results, the predicted score of social adaptability in the AI group is 0.266 while the non-AI group is 0.152 (see Table 2).

Then, we conducted an independent sample *t*-test and found that there were significant differences in social adaptability between the *AI* group and the *non-AI* group (Figure 2). As is shown in Table 3, the score of social adaptability of the *AI* group (*M* ± *SD* =5.42 ± 14.36) is significantly higher than that of the *non-AI* group (*M* ± *SD* =2.84 ± 13.87, *t*(1326) = 2.799, *p* < 0.01; see Table 4), indicating that AIEd, instead of having negative effect on the development of adolescents’ social adaptability, it may even promote the development of adolescents’ social adaptability to some degree. This finding is out of our expectations, and also demonstrates that the influence of AIEd on adolescents’ social adaptability is quite different from other products of information technology.

We further analyzed the relation between “the reason for students to use AI” and their social adaptability. From the results, we found that the proportion of students who choose AI for learning due to school assignments is the highest (43.00%), followed by students who choose AI for learning due to their interest (28.60%). Additionally, the proportion of students who choose AI for learning due to both their interest and school assignments is 14.10%.

The differences in social adaptability among students who use AI for different reasons are significant (*F* = 2.910, *p* < 0.01). Among them, the score of social adaptability of students who use AI because of their own interests as well as meet the requirements of their parents was the highest (*M* = 11.63 ± 11.07) while students who use AI to meet the requirements of their parents and school assignments is lower (*M* = 10.69 ± 14.74). It is worth mentioning that students who use AI to learn in order to meet the request of their parents obtained the lowest scores on social adaptability (*M* = 3.25 ± 14.59).

Then, we also further investigated the attitude of students toward AI. From the Table 5, we easily found that most of the students think that learning with AI can bring benefits (89.30%), and only a small number of students think that learning with AI can bring no benefit (10.70%). The results also showed that there was a significant difference in social adaptability between two groups of students who hold different attitudes toward AI (*t* = 2.410, *p* < 0.05; see Table 6). The scores of social adaptability of students who consider that AI is beneficial (*M* = 5.21 ± 14.29) were significantly higher than those who thought that AI was unhelpful (*M* = 1.74 ± 13.14).

From the Table 7 and Table 8, we can conclude that both individual factors and interpersonal factors, as well as social factors, may play crucial roles in the development of adolescents’ social adaptability. That is, there may be several factors contributing to the results. As to individual factors, utilizing AI to learn may improve learning efficiency so that students can spend more time on social activities, which may also improve their social adaptability. What is more, in the aspect of social factors, AI may provide a more effective way for people to communicate [41]. For example, students can use AIEd to give feedback to their teacher more freely. It may, to some degree, improve students’ communication skills.

Therefore, in order to understand why AIEd can promote the development of adolescents’ social adaptability, we conducted further research on the possible influences of the psychosocial variables on social adaptability among AI group students.

### 3.2. Descriptive Analysis of Several Psychosocial Factors

From the results of the overall average, the scores of interpersonal relations, teacher–student relations, interparental relations, peer relations, and the empathy ability of these students are higher than the average of the scale, indicating that the AI group students are better in all kinds of interpersonal relationships and empathy in real life (see Table 9). The scores of cognitive reappraisal strategy and expression inhibition strategy are also higher than the average of the scale, indicating that students often use these two strategies in dealing with their emotions. The results of academic emotion show that the score of students’ academic boredom is lower than the average value of the scale, and the score of academic anxiety and academic negative emotion is slightly higher than the average value of the scale, indicating that students have less boredom about learning, but they will be more anxious about learning and have more negative emotions when studying. The results of loneliness and impulsivity show that the scores of both are lower than the average of the scale, indicating that the overall level of loneliness and impulsivity of students is low. In addition, the score of social adaptability of this group is also lower on the whole.

### 3.3. Correlation between Psychosocial Factors and Social Adaptability

The figure shows the correlation between variables and social adaptability (see Table 10). According to the results, the interpersonal factors (interpersonal relations, teacher–student relations, interparental relations, peer relations) and individual factors (empathy, emotion regulation strategies, academic emotion, loneliness, impulsivity) investigated are related to social adaptability to varying degrees. Among them, interpersonal relations (*r* = 0.467, *p* < 0.01), teacher–student relations (*r* = 0.259, *p* < 0.01), interparental relations (*r* = 0.336, *p* < 0.01), peer relations (*r* = 0.434, *p* < 0.01), empathy (*r* = 0.112, *p* < 0.01), cognitive reappraisal strategy in emotion regulation strategy (*r* = 0.254, *p* < 0.01) were positively correlated with social adaptive ability, while the expression inhibition strategy in emotion regulation strategies (*r* = −0.073, *p* < 0.01), academic emotion (*r* = −0.445, *p* < 0.01), academic anxiety (*r* = −0.333, *p* < 0.01), academic boredom (*r* = −0.381, *p* < 0.01), loneliness (*r* = −0.446, *p* < 0.01) and impulsivity (*r* = −0.137, *p* < 0.01) were negatively correlated with social adaptability.

### 3.4. Psychosocial Factors That Predict the Adolescents’ Social Adaptability

To further investigate the possible influences of the variables on social adaptability among AI group students, mean square error (MSE) was adopted to measure the importance of these psychosocial variables. The higher value of Inc MSE means that the factors are more important. We can easily find that the four most important factors are interpersonal relationships, peer relations, academic emotion, and loneliness, as seen in Figure 3. This finding is consistent with our expectation that both social environmental factors and individual factors play crucial roles in predicting adolescents’ social adaptability. To be specific, adolescents with a high level of interpersonal relations are more likely to exhibit a good status of social adaptability. During the epidemic period, AIEd may provide an effective way of communication so that adolescents that can maintain good relationships between peers or teachers. What is more, AIEd may also provide a way for adolescents to prevent feeling anxiety and loneliness when they are finishing academic tasks, especially collaborative tasks.

In addition, in order to conduct a test of the significance of the importance, the permutation test was adopted. As is shown in Figure 3, all the variables can significantly predict the status of social adaptability except for gender, expression inhibition strategy, and impulsivity.

## 4. Conclusions

The current study examined whether AIEd may influence adolescents’ social adaptability and whether these psychosocial factors exactly predict the status of adolescents’ social adaptability among *AI* group students. The machine learning method was also utilized in the present study. Additionally, we find a positive influence of AIEd on adolescents’ social adaptability.

The negative social relationship had a significant influence on adolescents’ social adaptability. Specifically, poor interpersonal relationships and peer relations may cause a lack of social attachment that may influence the development of social adaptability among AI group students. In addition, loneliness and academic emotion also play important role in predicting adolescents’ social adaptability. These findings emphasize the significant impact of social environment factors, interpersonal factors, and learner-related factors on adolescents’ social adaptability.

However, this study featured a cross-sectional design, meaning it was unable to make causal inferences. Future studies can adopt longitudinal studies to further investigate the relation between AIEd and adolescents’ social adaptability. In addition, a clear definition of AIEd, as well as a full discussion about man–machine relationships (Li, 2020), should be investigated in the future study.

## Figures and Tables

**Figure 1 ijerph-19-07890-f001:**
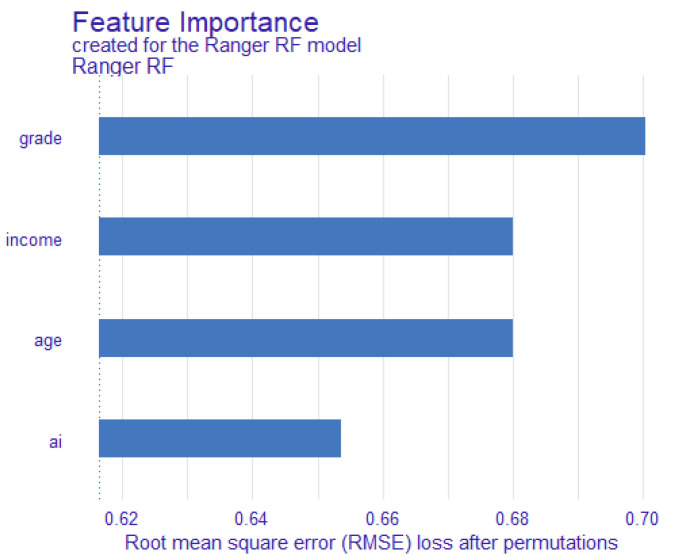
Importance ranking of variables on social adaptability predicted by RF.

**Figure 2 ijerph-19-07890-f002:**
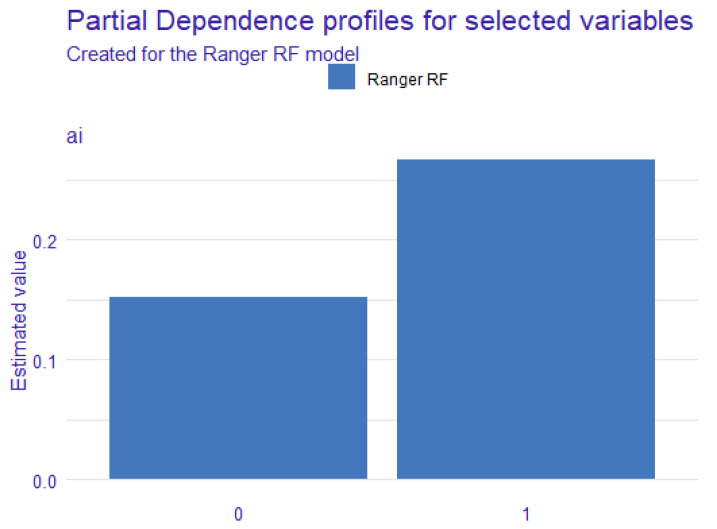
Social adaptability differences between AI group and non-AI group predicted by PDP.

**Figure 3 ijerph-19-07890-f003:**
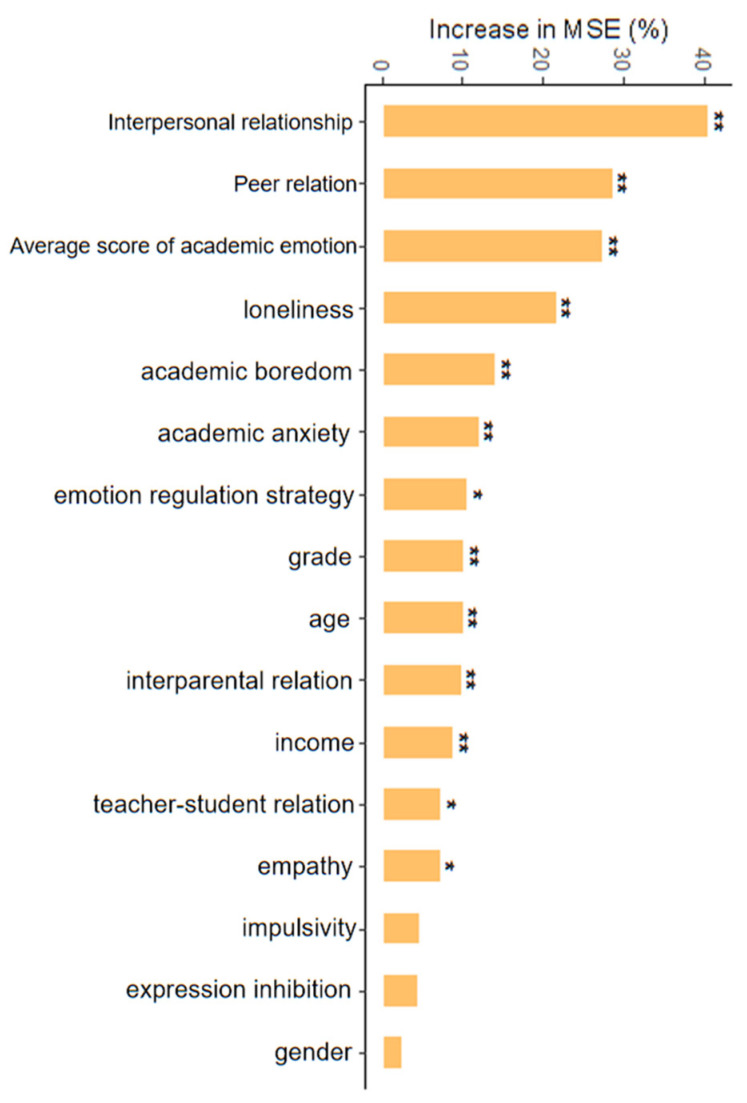
Importance ranking of variables on social adaptability predicted by MSE. * = *p* < 0.05, ** = *p* < 0.01.

**Table 1 ijerph-19-07890-t001:** Details of “whether to use AI for learning”.

Group	Number			Proportion
AI group	1016	Male	516	38.86%
		Female	500	37.65%
Non-AI group	312	Male	189	14.23%
		Female	123	9.26%
Total	1328			100%

**Table 2 ijerph-19-07890-t002:** Social adaptability differences between AI group and non-AI group.

	Model	Label X	Y
1	AI Ranger RF	0 (Non-AI group)	0.152
2	AI Ranger RF	1 (AI group)	0.266

Group was dummy-coded such that 1 = AI group, 0 = non-AI group.

**Table 3 ijerph-19-07890-t003:** Results of differences in social adaptability between *AI* group and *non-AI* group.

Group	Social Adaptability				
	*N*	*M*	*SD*	*t*	*p*
AI group	1016	5.42	14.36	2.799 **	0.005
Non-AI group	312	2.84	13.87		

** = *p* < 0.01.

**Table 4 ijerph-19-07890-t004:** Social adaptability of AI group and non-AI group.

Group	Social Adaptability				
	*N*	*M*	*SD*	*t*	*p*
AI group	1016	5.42	14.36	2.799 **	0.005
Non-AI group	312	2.84	13.87		

** = *p* < 0.01.

**Table 5 ijerph-19-07890-t005:** Details of students’ choice on “Why use AI for learning”.

Group	Frequency	Proportion (%)	Cumulative Proportion (%)
Personal interest	291	28.60	28.60
Parents’ Request	33	3.20	31.90
School Arrangements	437	43.00	74.90
Personal interest and Parents’ Request	43	4.20	79.10
Personal interest and School Arrangements	143	14.10	93.20
Parents’ Request and School Arrangements	14	1.40	94.60
Personal interest and Parents’ Request and School Arrangements	55	5.40	100.00
Total	1016	100.00	

**Table 6 ijerph-19-07890-t006:** Analysis results of social adaptability scores of students among different AI learning cause groups.

Group	Social Adaptability				
	*N*	*M*	*SD*	*F*	*p*
Personal interest	291	5.48	14.13	2.910 **	0.008
Parents’ Request	33	3.25	14.59		
School Arrangements	437	3.65	14.07		
Personal interest and Parents’ Request	43	11.63	11.07		
Personal interest and School Arrangements	143	4.24	15.08		
Parents’ Request and School Arrangements	14	10.69	14.74		
Personal interest and Parents’ Request and School Arrangements	55	6.52	13.67		

** = *p* < 0.01.

**Table 7 ijerph-19-07890-t007:** Details of students’ choice on the question “I think it is helpful for me to use AI to learning”.

Group	Frequency	Proportion (%)	Cumulative Proportion (%)
Yes	907	89.30	89.30
No	109	10.70	100.00
Total	1016	100.00	

**Table 8 ijerph-19-07890-t008:** Results of the difference in scores of social adaptability between the YES group and the NO group.

Group	Social Adaptability				
	*N*	*M*	*SD*	*t*	*p*
Yes	907	5.21	14.29	2.410 *	0.016
No	109	1.74	13.14		

* = *p* < 0.05.

**Table 9 ijerph-19-07890-t009:** Mean and standard deviation of variables.

Variable	SA	IR	TSR	IPR	PR	EMP	CR	EI	AE	AA	AB	LON	IMP
*M*	4.81	0.71	4.07	3.52	4.36	3.58	4.81	4.40	2.60	3.30	2.16	2.05	2.33
*SD*	0.71	0.22	0.76	0.76	0.68	0.51	1.24	1.27	0.75	0.99	0.89	0.53	0.34

SA = social adaptability, IR = interpersonal relation, TSR = teacher–student relation, IPR = interparental relation, PR = peer relation, EMP = Empathy, CR = cognitive reappraisal strategy, EI = expression inhibition strategy, AE = academic emotion, AA = academic anxiety, AB = academic boredom, LON = Loneliness, IMP = Impulsivity.

**Table 10 ijerph-19-07890-t010:** Correlation results among variables.

	1	2	3	4	5	6	7	8	9	10	11	12	13
SA	1												
IR	0.467 **	1											
TSR	0.259 **	0.238 **	1										
IPR	0.336 **	0314 **	0.312 **	1									
PR	0.434 **	0.381 **	0.313 **	0.282 **	1								
EMP	0.112 **	0.049	0.251 **	0.173 **	0.346 **	1							
CR	0.254 **	0.142 **	0.223 **	0.234 **	0.276 **	0.202 **	1						
EI	−0.073 **	−0.121 **	0.059 *	0.038	0.000	−0.032	0.441 **	1					
AE	−0.445 **	−0.393 **	−0.221 **	−0.283 **	−0.228 **	−0.025	−0.153 **	0.020	1				
AA	−0.333 **	−0.356 **	−0.065 *	−0.163 **	−0.80 **	0.154 **	−0.051	0.074 **	0.724 **	1			
AB	−0.381 **	−0.292 **	−0.260 *	−0.277 **	−0.260 **	−0.144 **	−0.175 **	−0.025	0.873 **	0.295 **	1		
LON	−0.446 **	−0.452 **	−0.288 **	−0.380 **	−0.450 **	−0.222 **	−0.180 **	0.057 *	0.458 **	0.273 **	0.441 **	1	
IMP	−0.137 **	−0144 **	−0.105 **	−0.100 **	−0.052	−0.048	−0.044	−0.049	0.348 **	0.181 **	0.354 **	0.292 **	1

* = *p* < 0.05, ** = *p* < 0.01, SA = social adaptability, IR = interpersonal relation, TSR = teacher–student relation, IPR= interparental relation, PR = peer relation, EMP = Empathy, CR = cognitive reappraisal strategy, EI= expression inhibition strategy, AE = academic emotion, AA = academic anxiety, AB = academic boredom, LON = Loneliness, IMP = Impulsivity.

## Data Availability

The data that support the findings of this study are available from the corresponding author, upon reasonable request.

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
