# Peer review of "Influence of Artificial Intelligence in Education on Adolescents’ Social Adaptability: A Machine Learning Study"

_ijerph, 2022, doi:10.3390/ijerph19137890_

Round 1

Reviewer 1 Report

I feel that the authors have satisfactorily dealt with my previous comments. The formatting, however, needs to be checked as there are differences (for example in size) in the paper.

Author Response

Thanks for raising this point. We have revised the format of our manuscript according to the MDPI rules. Besides, revisions made to the manuscript were marked up using the “Track Changes” function.

Reviewer 2 Report

I have now read the revised version and I believe the manuscript has been improved and could be published.

Author Response

Thank you for your previous suggestions for our manuscript. It contributed a lot to our study and future research.

Reviewer 3 Report

Thank you for the opportunity to review this work. The paper addresses a very interesting topic, that is, the influence of Artificial Intelligence in education on adolescents’ social adaptability. To do this, the paper uses quantitative research methods, through a questionnaire in several schools in Guangdong (China).

The paper’s argument is built on an appropriate base of theory, and the research is well designed.  In this case the statement of objectives such as identify the relevant psychosocial factors that can predict adolescents’ social adaptability, between others, are correct for the resolution of the hypothesis. The introduction covers a lot of background material, it demonstrates an adequate understanding of the relevant literature in the field and cite an appropriate range of literature sources. The first paragraphs point out a very important idea.

Some minor changes:

- Keywords are missing, must be included.

- Some paragraphs are in the wrong font.

- A revision of the English of the text must be done since some sentences are complex to understand.

As a final assessment, the article is written correctly from the methodical point of view. The formal and substantive assessment is positive, although it would be good to address all the comments submitted. After taking them into account, the article will be in perfect condition for publication.

Author Response

  1. Keywords are missing, must be included.

 We appreciate this suggestion and the keywords were also added according to the MDPI rules.

  1. Some paragraphs are in the wrong font.

We appreciated this suggestion and have corrected the format of our manuscript according to the MDPI rules.

  1. A revision of the English of the text must be done since some sentences are complex to understand.

Thanks for raising this point. We have rewritten some sentences according to the suggestions.

This manuscript is a resubmission of an earlier submission. The following is a list of the peer review reports and author responses from that submission.

Round 1

Reviewer 1 Report

The study investigates the influence of Artificial Intelligence in Education and psychosocial factors that might influence/predict adolescents' social adaptability.

The introduction is satisfactory, although I would suggest to the authors to describe in more detail what is social adaptability and to report previous research on the topic.

What I am most concerned of is the analysis plan. I will describe below what I think are the major flaws:

  • it is not very clear how the AI present vs. AI absent groups were defined. Also, there is no indication on how many participants belonged to each group. Yet this is a very relevant point.
  • The authors write about the artificial intelligence usage questionnaire, but it is not clear if and how it was used in the analyses. Further, the scores on this questionnaire are an interesting and potentially relevant predictor of social adaptability
  • it is not clear which of the several questionnaires that have been used had already been validated and which was not. It seems, for example, that the following were not as there is no reference for them. Further, the author write about reliability and validity, the latter of which cannot be explored until factor analysis and criterion/construct validity analyses are conducted (which I would suggest the authors to do)
    • Artificial intelligence usage questionnaire, for which the alpha is missing
    • Social adaptability
    • Intraparental relation
    • Teacher-student relation
    • Academic emotion
  • there is the need to define in more detail what is the difference between the ai and the non ai groups
  • I am also concerned regarding the analysis plan. It is not clear why the authors opted for a machine learning algorithm when other analyses, perhaps more effective, can be conducted. It is also not clear why, after the machine learning algorithm has been adopted, the following analyses (MSE) are conducted only on the ai-present group. Considering the aims of the study, I would suggest the authors to revise the analysis strategy. For example, a hierarchical regression with 2 or 3 different blocks (1st, socio-demographic data; 2nd, dummy variable for ai present vs ai absent; 3rd psychosocial factors) can be conducted
  • the conclusion is too short, and it does not relate to previous research nor it discusses limitations and future directions in detail
  • last, English language needs a cautious check as there are several typos and errors and some sections of the paper are not very clear

Reviewer 2 Report

Authors report a study to investigate the influence of (AIEd) on adolescents’ social adaptability, a machine learning algorithm was used to find the relevant psychosocial variables. Results showed that the four most important factors in the prediction of social adaptability are interpersonal relationships, peer relation, academic emotion, and loneliness.

This is timely research; some changes are suggested:

  • I appreciate the introduction is well written and uses up to date references. I think the last point where the hypotheses and purpose are introduced should be extended, the hypotheses shouldn’t be into a paragraph and should be linked to the previous literature.
  • The sample should be more detailed
  • What is the ethics committee reference?
  • The same aspect in me measures than in the sample, it includes a lot of text, but that information should be provided in a visual format to better understand the methodology than in “texty” paragraphs.
  • Results are well present and clear to me, but the paper lacks a discussion to understand the impact of the study, what are the implications of the study, and the next steps for research?

Reviewer 3 Report

The manuscript entitled ‘Influence of Artificial Intelligence in Education on Adolescents’ Social Adaptability: A Machine Learning Study’ investigates the influence of Artificial Intelligence in education (AIEd) on adolescents’ social adaptability. 
Introduction: clear, the gap in the literature and the rationale for the study is explained well. 

Methods: needs some improvement. The context of the study needs to be described better, with more details. For example, from which type of schools (e.g. Primary, secondary, etc.) and how many schools the data were collected should be given in the methods section. 
2.2. measures: should be changed as ‘data collection instruments’. As it was given in the manuscript there were 10 different instruments used to collect the data. Is that correct? If it is so, there should be very rich data which can lead to rich results. However, this is not remarkable in the results section. 
In the explanation of ‘peer relation scale’ it is given that it is ‘six point scale with 1= strongly agree and 5= strongly disagree. So, it should be a ‘five point scale’. 
The 10 data collection instruments includes totally 129 items. How much time was given to the students to complete them? 
Did the researchers administered all the instruments at once? 
The procedure of the data collection and the procedure of the study should have been explained with more details. So, the readers can understand how this study was done. 
From the results of the study, it is not clear what is the value of using machine learning to investigate the influence of Artificial Intelligence in education (AIEd) on adolescents’ social adaptability. In the results section it is given that by using a t-test the researchers have already investigated this. 
The data analysis section is missing in the manuscript. There should be a separate section to explain how the data were analysed. The data analysis and the results are presented together in the manuscript which should be corrected. In the data analysis section it should be explained why certain statistical methods and tests were used. 
The ‘discussion’ section is missing. The authors should discuss the findings of the study in a separate section ‘discussion’. 
What are the implications of the findings of this study in education practice? Should also be added. 
Not all the figures were explained and discussed. The discussion of the results that were presented in figure should be added. 
There are some English language mistakes. Such as ‘how many times….’ (Page 4, line 1) which should be ‘how much time….’. So, the manuscript needs a proofreading.